# Roles of mTOR in the Regulation of Pancreatic β-Cell Mass and Insulin Secretion

**DOI:** 10.3390/biom12050614

**Published:** 2022-04-21

**Authors:** Shun-ichiro Asahara, Hiroyuki Inoue, Hitoshi Watanabe, Yoshiaki Kido

**Affiliations:** 1Division of Diabetes and Endocrinology, Department of Internal Medicine, Kobe University Graduate School of Medicine, Kobe 6500017, Japan; 2Division of Medical Chemistry, Department of Metabolism and Diseases, Kobe University Graduate School of Health Sciences, Kobe 6540142, Japan; h.inoue310@gmail.com (H.I.); kido@med.kobe-u.ac.jp (Y.K.); 3Department of Medicine, Vagelos College of Physicians and Surgeons, Columbia University, New York, NY 10032, USA; hw2775@cumc.columbia.edu; 4Naomi Berrie Diabetes Center, Vagelos College of Physicians and Surgeons, Columbia University, New York, NY 10032, USA

**Keywords:** mTOR, pancreatic β-cell, autophagy, ER stress, insulin secretion

## Abstract

Pancreatic β-cells are the only type of cells that can control glycemic levels via insulin secretion. Thus, to explore the mechanisms underlying pancreatic β-cell failure, many reports have clarified the roles of important molecules, such as the mechanistic target of rapamycin (mTOR), which is a central regulator of metabolic and nutrient cues. Studies have uncovered the roles of mTOR in the function of β-cells and the progression of diabetes, and they suggest that mTOR has both positive and negative effects on pancreatic β-cells in the development of diabetes.

## 1. Introduction

The number of diabetic patients continues to increase worldwide, causing a variety of problems [1]. The development and progression of diabetes can lead to microvascular complications, including neuropathy, nephropathy, and retinopathy [2]. In addition, the risk of macrovascular complications, such as cardiovascular disease and cerebrovascular disease, has become a major issue. Although the risk of dying from these diabetic complications has been decreasing in recent years, they still threaten the lives of many patients worldwide and reduce their quality of life [3,4].

The majority of diabetic patients are diagnosed with type 2 diabetes mellitus, which can be divided into two major pathologies: insulin resistance, in which insulin becomes less effective, and decreased insulin secretion from pancreatic β-cells [5]. Insulin resistance means that insulin does not play its normal role in target organs, such as the liver, skeletal muscle, central nervous system, and adipose tissue, despite the presence of insulin in the blood. In contrast, insulin hyposecretion is caused by a decrease in pancreatic β-cell mass, the only tissue that secretes insulin, or by an abnormality in the insulin secretory mechanism. The pathway shared by these mechanisms is the insulin signaling pathway, which is activated by the binding of insulin to insulin receptors, and it plays a role in promoting cell proliferation and growth [6]. Insulin also inhibits glycogenesis in the liver, promotes glucose uptake in skeletal muscle, and suppresses lipolysis in adipocytes [7,8,9]. Insulin resistance develops when there is some disturbance in the activation of these signals. Insulin signaling is also important for the regulation of pancreatic β-cell mass [10,11,12,13], and it regulates glucose-stimulated insulin secretion by pancreatic β-cells [14]. That is, the disruption of insulin signaling in pancreatic β-cells results in a decrease in pancreatic β-cell mass and a decrease in insulin secretory capacity. The impairment of insulin signaling can be attributed to a variety of causes, including genetic factors, environmental factors, and gene–environment interactions [15]. At the center of these causes is nutrition. Given that overnutrition is a trigger for the development of diabetes, the effect of nutrition on insulin signaling seems to be significant. The mammalian target of rapamycin (mTOR) is an important molecule in the insulin signaling pathway. mTOR plays a central role in sensing extracellular nutrient status, intracellular energy, and other information, and it links them to cell growth and proliferation [16,17,18]. It is also well known that mTOR is an oncogene and has been clinically applied as a target molecule for cancer therapy [19,20]. Various roles of mTOR in glucose metabolism have been reported in skeletal muscle, kidney, and liver [21,22,23], and particularly interesting findings have been published regarding its action in pancreatic β-cells. As mentioned above, the “mass” and “function” of pancreatic β-cells are important for the maintenance of blood glucose levels, and mTOR is involved in both processes [24,25,26,27,28]. At the same time, however, it remains controversial whether mTOR provides a benefit or a risk to pancreatic β-cells [29]. Therefore, in this review, we introduce reports on the role of mTOR in pancreatic β-cells and focus on the positive and negative effects of mTOR on them.

## 2. Basic Knowledge of mTOR

mTOR is a serine/threonine kinase that plays a central role in sensing extracellular nutrient status and intracellular ATP levels to promote cell growth and proliferation [16,17,18]. mTOR exists in two large protein complexes within the cell, namely, mTOR complexes 1 and 2 (mTORC1 and mTORC2), and both complexes are activated by signals from growth factors, such as insulin [30] (Figure 1).

mTORC1 is composed of mTOR; regulatory associated protein of mTOR complex 1 (RAPTOR); proline-rich AKT1 substrate; 40 kDa (PRAS40); DEP domain-containing mTOR-interacting protein (DEPTOR); and mTOR-associated protein, LST8 homolog (mLST8). mTORC1 is activated by growth factors, such as insulin and insulin-like growth factor 1, via the phosphorylation of TSC complex subunit 2 (TSC2) by AKT. TSC2 is stabilized in the intracellular TSC1-TSC2 complex, which is a GTPase-activating protein (GAP) of Ras homolog enriched in brain (Rheb). However, when TSC2 activity is inhibited by upstream signals, it activates mTORC1 via Rheb activation [31]. Activated mTORC1 phosphorylates S6 kinase (S6K1) and eukaryotic translation-initiation factor 4E binding protein 1 (4E-BP1), which, in turn, phosphorylate various ribosomal proteins and promote translation. 4E-BP1 binds to mRNA and represses translation, but when phosphorylated, it separates from mRNA and initiates translation. That is, mTORC1 plays a central role in translational regulation [32]. In addition, mTORC1 inhibits autophagy by phosphorylating autophagy-related 13 (ATG13) and Unc-51-like autophagy-activating kinase 1 (ULK1), which are responsible for this phenomenon [33,34]. Furthermore, mTORC1 is involved in the regulation of lipid synthesis and mitochondrial biogenesis [35,36].

mTORC2 is composed of DEPTOR and mLST8, which are both in mTORC1; mTORC2-specific RAPTOR-independent companion of mTOR complex 2 (RICTOR); mammalian stress-activated protein kinase-interacting protein 1 (mSIN1); and protein observed with RICTOR (PROTOR). The activation mechanism of mTORC2 is less well understood compared with that of mTORC1, but mTORC2 is reported to be activated by growth factors, such as insulin [17,37]. In addition, mTORC2 enhances the activity of AKT by phosphorylating serine 473 of AKT. In particular, it plays a significant role in the regulation of the cytoskeleton [16,17].

mTORC1 is activated by extracellular nutritional conditions, including amino acid levels. The stimulation of cells with amino acids activates mTORC1 by transferring mTORC1 to the lysosomal membrane via the Rag GTPase complex [38,39]. Glycolysis and mitochondrial oxidative phosphorylation suppress AMPK activity by increasing the intracellular ATP/AMP ratio, thereby activating mTORC1 [40]. In contrast, during energy deprivation, such as fasting, AMPK is activated by a decrease in the ATP/AMP ratio, resulting in the suppression of mTORC1 activity [41,42].

## 3. mTOR and Insulin Secretion

mTORC1 activation enhances protein translation, which also promotes insulin synthesis [43,44]. Mitochondria that produce ATP, which is required for insulin secretion, are also regulated by mTORC1 activity [36].

The subunits of the respiratory chain complexes (I, III, IV, and V) are made of 13 proteins encoded by mitochondrial DNA. The complexes produce most of the energy required for cellular activity [45]. Mitochondrial diabetes is caused by pancreatic β-cell failure resulting from mutations in mitochondrial DNA [46]. The transcription factor peroxisome proliferator-activated receptor gamma coactivator 1-alpha (PGC1α) is a regulator of mitochondrial biogenesis [47,48]. PGC1α is a coactivator of the transcription factor nuclear respiratory factor (NRF)-1/2, which activates transcription factor A, mitochondrial (TFAM), thereby inducing the transcription and stabilization of mitochondrial DNA [49,50,51]. Pancreatic β-cell-specific TFAM-knockout mice show reduced insulin secretory capacity in association with reduced mitochondrial DNA content and abnormal mitochondrial morphology [52]. Although AMP-activated protein kinase (AMPK) is a key molecule in the regulation of insulin secretion and pancreatic β-cell mass [53], it is also known to be a significant factor in mitochondrial biogenesis [54]. Aminoimidazole carboxamide ribonucleotide, which activates AMPK, promotes mitochondrial biosynthesis via PGC1α and NRF [55,56]. It has been shown that mTOR enhances mitochondrial function in the HEK293 cell line [57]. Furthermore, rapamycin inhibits mitochondrial gene transcription by dissociating PGC1α from the complex of mTORC1 and the transcription factor YY1 [58]. YY1 functions as a coactivator of PGC1α, and the YY1-PGC1α complex is important for mitochondrial gene transcription, but its function is dependent on mTORC1 activity. In the skeletal muscle of type 2 diabetic patients, mitochondrial density and protein levels are decreased, but the expression levels of PGC1α, NRF, and TFAM remain unchanged [59], suggesting that mitochondrial gene expression is regulated by the extracellular nutritional environment and growth factors via mTORC1. PGC1α activation is known to be affected not only by its expression level but also by protein modifications, such as deacetylation and phosphorylation [60]. PGC1α deacetylation is mediated by sirtuin 1 and phosphorylated AMPK, and as a result, PGC1α acts as a coactivator of transcription factors [61,62]. Because TSC2 deficiency and mTORC1 activation lead to AMPK phosphorylation [63,64], it is possible that increased levels of phosphorylated AMPK deacetylate PGC1α and increase its activity [65] (Figure 2).

Statins are widely used cholesterol-lowering drugs, but their administration is reported to cause decreased insulin secretion and hyperglycemia [66,67]. Type 2 diabetes and hyperlipidemia often coexist, and their side effects are a major problem. However, the mechanism by which statins decrease insulin secretion is not well understood, although mTORC1 has been implicated in this process [68]. High-fat-diet-fed mice treated with the statin atorvastatin have decreased insulin secretion and insulin granules. Transcriptome profiling of islets from these mice showed the decreased expression of various transcription factors and decreased mTOR signaling. Rab5a, a small G protein, was downregulated by atorvastatin, suggesting that Rab5a positively regulates mTORC1 activity in pancreatic β-cells.

Although the palmitic acid loading of pancreatic islets induces compensatory hyperplasia and insulin hypersecretion in the acute phase and pancreatic β-cell failure in the chronic phase, the mechanism is not well understood [69,70,71]. Hatanaka et al. found that palmitate loading increases the polyribosomal occupancy of total RNA and increases mRNA translation [72]. This translation-promoting effect was due to the activation of the mTOR pathway via L-type Ca^2+^ channels and was independent of insulin signaling. At longer incubation times, the levels of polyribosome-associated RNA are decreased, leading to the activation of the unfolded protein response (UPR).

## 4. mTORC1 and Regulation of Pancreatic β-Cell Mass

Many reports have shown that mTORC1 influences pancreatic β-cell mass because it is involved in cell proliferation and growth. Factors that suppress mTORC1 include TSC1 and TSC2 complexes, and mice with pancreatic β-cell-specific activation of mTORC1 have been generated and analyzed by deleting these genes [44,73,74,75,76,77,78] (Figure 3). Mice overexpressing Rheb, a target molecule of TSC2, specifically in pancreatic β-cells have also been generated, and mTORC1 activation is observed in the islets of these mice [79]. Rheb-overexpressing mice show an increase in pancreatic β-cell mass and a marked enhancement of insulin secretion, resulting in an improvement in glucose tolerance. In addition, islets isolated from pancreatic β-cell-specific mTOR-knockout mice have abnormal mitochondrial function and decreased insulin secretion due to oxidative stress. Gene expression analysis of the pancreatic islets of these mice revealed the increased expression of thioredoxin-interacting protein (TXNIP) and carbohydrate-responsive element-binding protein (ChREBP), consistent with the results from the islets of diabetic model mice and islets from type 2 diabetic patients. In contrast, the binding of mTOR to the ChREBP-Max-like protein complex reduces its transcriptional activity and decreases TXNIP expression, thereby suppressing oxidative stress and apoptosis [80,81]. Furthermore, the inhibition of mTORC1 activity by deleting RAPTOR specifically in mouse pancreatic β-cells is accompanied by increased apoptosis of postnatal pancreatic β-cells, as well as impaired glucose-stimulated insulin secretion and reduced β-cell mass [82]. In addition, mice overexpressing a pancreatic β-cell-specific mTOR kinase-dead mutant show a normal proliferative trend in β-cell mass but become glucose intolerant due to abnormal insulin secretion caused by a deficiency of pancreatic and duodenal homeobox 1 (PDX1) [83]. These results indicate that mTOR is essential for the maintenance of normal pancreatic β-cell mass and insulin secretion.

Recent reports indicate that mTORC1 activity plays an important role in pancreatic β-cell growth during embryonic and neonatal periods. The inhibition of mTORC1 activity in fetal pancreatic β-cells affects the growth and differentiation of pancreatic endocrine cells, resulting in hyperglycemia in the neonatal period [85]. When pregnant mice are fed a low-protein diet, mTORC1 activity is decreased through changes in microRNA expression during fetal development, resulting in impaired insulin secretion, decreased PDX1 expression, and decreased pancreatic β-cell mass [88]. Furthermore, mice lacking S6K1 exhibit intrauterine growth restriction (IUGR) and reduced pancreatic β-cell mass, but the restoration of IUGR by tetraploid embryo complementation does not improve pancreatic β-cell mass [89]. These results suggest that S6K1 regulates pancreatic β-cell mass independently of IUGR. In humans, fetal growth restriction due to placental abnormalities of nutrient transport is associated with decreased S6K1 phosphorylation [90], but S6K1 phosphorylation is increased in the placenta of patients with gestational diabetes mellitus [91], which may be a compensatory response to promote fetal growth, as well as one of the mechanisms by which infants born with gestational diabetes mellitus become gigantic. Recently, cell signaling in pancreatic β-cells was reported to contribute to the maturation of pancreatic β-cells by switching from mTORC1 to AMPK from the fetal stage to the postnatal period [92]. When mTORC1 is homeostatically activated in post-mature pancreatic β-cells, the cells exhibit an immature phenotype. Thus, mTORC1 activity in pancreatic β-cells during embryogenesis and birth affects various aspects of pancreatic β-cell proliferation, apoptosis, and differentiation, and it is thought to be involved in pancreatic β-cell failure after growth.

## 5. Autophagy

Autophagy is an autolytic catabolic process that occurs within cells and is required for β-cell survival, insulin secretion, and blood glucose homeostasis. Electron microscopic analysis showed the abnormal accumulation of autophagosomes in MIN6 cells, a mouse insulin-secreting cell line, loaded with high free fatty acids or high glucose and in human islets from type 2 diabetic patients [93,94]. These results suggest that autophagy is abnormally regulated in type 2 diabetes, but whether autophagy in pancreatic β-cells is promoted or inhibited in diabetes is still controversial. However, there is no doubt that the regulatory mechanism of autophagy in pancreatic β-cells plays a crucial role in the development of type 2 diabetes and pancreatic β-cell failure. In particular, it has been shown that when mice lacking the autophagy-related gene Atg7 are fed a high-fat diet, β-cell apoptosis is increased, insulin secretion is decreased, compensatory β-cell hyperplasia is lost, and diabetes is enhanced [95]. mTORC1 is a negative regulator of autophagy and may affect β-cell function and survival via the suppression of autophagy in the type 2 diabetic state [93,96,97,98]. mTORC1 activation by glucotoxicity and lipotoxicity associated with diabetes, as well as genetic activation, induces the accumulation of p62 and impairs autophagy [94]. The constitutive activation of mTORC1 also impairs mitophagy, the autophagic removal of damaged mitochondria. Pancreatic β-cells from aged pancreatic β-cell-specific TSC2-knockout mice tend to have more degenerated mitochondria, which can lead to the depolarization of the mitochondrial membrane and increased oxidative stress, thereby causing apoptosis [99]. Furthermore, the restoration of autophagy via mTORC1 inhibition following treatment with rapamycin leads to the protection of β-cells [94]. mTORC1 inactivation by rapamycin improves β-cell function and blood glucose levels in Akita mice, a model of endoplasmic reticulum (ER) stress-induced diabetes, by enhancing autophagy [100]. These results indicate that autophagy and ER stress are strongly linked through proinsulin misfolding in pancreatic β-cells and that mTOR has roles in both processes. The relationship between ER stress and mTOR is discussed in the next section. Nevertheless, these observations suggest that inhibiting mTORC1 activity, which induces autophagy, may protect pancreatic β-cells. ULK1, an autophagy-initiating kinase, is a common substrate of AMPK and mTORC1. The phosphorylation of ULK1 (Ser 317 and Ser 777) by AMPK induces autophagy, and the phosphorylation of ULK1 (Ser 757) by mTORC1 dissociates the binding of ULK1 to AMPK; that is, mTOR and AMPK regulate autophagy via ULK1 under glucose stimulation, which, in turn, affects β-cell survival and insulin secretion [101]. It is thought that prolonged nutritional stress, such as type 2 diabetes, may inhibit AMPK activation in pancreatic β-cells, leading to chronic mTORC1 activation, which, in turn, impairs autophagy and mitophagy, resulting in pancreatic β-cell dysfunction and diabetes. Recently, Pasquier et al. reported macroautophagy-independent lysosomal degradation, termed stress-induced nascent granule degradation (SINGD) [102]. In the pancreatic β-cells of type 2 diabetic patients, SINGD is enhanced by the decreased expression of protein kinase D. Consequently, mTORC1 is recruited to the membrane of granule-containing lysosomes, and the chronic activation of mTORC1 inhibits macroautophagy. Thus, the aberrant activation of SINGD contributes to β-cell damage in type 2 diabetes [102]. In addition, it was recently shown that mTORC1 is regulated via Hippo signaling [86], which is an evolutionarily conserved pathway that regulates organ size by controlling apoptosis, cell proliferation, and stem cell self-renewal. In diabetic conditions, large-tumor suppressor 2 (LATS2), a core component of Hippo signaling, is activated to induce pancreatic β-cell apoptosis, which is also mediated by mTORC1. Activated LATS2 suppresses macroautophagy and induces pancreatic β-cell failure by homeostatically activating mTORC1. Thus, the mTORC1-mediated suppression of autophagy is regulated through various signals and is thought to be involved in the development of pancreatic β-cell failure in diabetic conditions.

## 6. ER Stress

The ER is an organelle that is responsible for protein folding in cells. Pancreatic β-cells, which are required to supply a large amount of insulin rapidly under insulin-resistant conditions, such as type 2 diabetes, accumulate a large number of unfolded proteins in the ER and are thus vulnerable to ER stress. ER stress triggers an adaptive response called the UPR to repair unfolded proteins and restore ER homeostasis to normal (adaptive UPR). However, if the imbalance between protein abundance and folding capacity persists and the UPR fails to restore ER function, chronic ER stress activates complex intracellular signaling pathways and triggers apoptosis via apoptosis-inducing factors, such as CHOP (terminal UPR) [103,104,105]. Since 2002, when ER stress was found to occur in pancreatic β-cells of type 2 diabetic patients, a chronic hyperglycemic load has been shown to be an important pathological factor in pancreatic β-cell failure, and many papers on the relationship between ER stress and pancreatic β-cell failure have been published [106]. The induction of ER stress in diabetes mellitus by hyperglycemia and hyperlipidemia is mediated by the activation of mTORC1 upon nutritional stimulation [107,108]. Furthermore, it is easy to imagine that many misfolded proteins accumulate in pancreatic β-cells because protein translation is enhanced by increased mTORC1 activity. Embryonic fibroblasts isolated from mice lacking pancreatic β-cell-specific TSC2 (βTSC2KO mice) exhibit severe ER stress and undergo apoptosis, suggesting that mTORC1 promotes ER stress. Islets isolated from these mice show increased expression of several UPR markers, including PRKR-like ER kinase (PERK), C/EBP-homologous protein (CHOP), activating transcription factor 4 (ATF4), and CCAAT enhancer-binding protein beta (C/EBPβ), indicating that mTORC1 promotes ER stress [73,99,109]. High glucose enhances palmitate-induced ER stress in INS-1E cells, a rat β-cell line, by activating the IRE1a-JNK pathway and promoting apoptosis in an mTORC1-dependent manner. This response has been confirmed by the fact that mTORC1 inhibition by rapamycin suppresses apoptosis by inhibiting X-box-binding protein 1 splicing, PERK phosphorylation, and CHOP expression [110]. The inhibition of mTORC1 was shown to alleviate ER stress in MIN6 cells exposed to lipophilic conditions [72]. In contrast, although ER stress is increased in neonatal pancreatic β-cells of Akita mice, temporarily increasing mTORC1 activity during this period is effective in maintaining pancreatic β-cell mass in adulthood [111].

The mechanism by which mTORC1 affects β-cell survival under ER stress has recently become clearer. Because mTORC1 promotes translation, it is possible that it also promotes apoptosis during ER stress due to the accumulation of additional misfolded proteins [112,113]. Furthermore, it has been shown that mTORC1 activation has roles in ATF4-induced translation recovery, increased amino acid flux, and protein synthesis [112]. mTORC1 also regulates ATF4 expression by stabilizing ATF4 mRNA and promoting its translation through 4E-BP1 [114]. That is, mTORC1 has been found to regulate ER stress at various stages of the UPR.

## 7. mTORC2 and β-Cell Growth and Function

As mentioned above, compared to mTORC1, there are many aspects of mTORC2 that are not well understood, both its upstream and downstream signals. However, there have been several reports on mTORC2-specific roles in pancreatic β-cells, which we discuss in this chapter.

Similar to mTORC1, mTORC2 is activated by growth factors and signals to several effectors. Insulin is best known to stimulate mTORC2 activity, which is mediated by PI3K, among other growth factors. The downstream effectors of mTORC2 include the PKC family, SGK1, and MST1, which promote insulin secretion and cell growth through the activation of these molecules [115]. mTORC2-specific components include RICTOR and mSIN1. To investigate the specific role of mTORC2 in pancreatic β-cells, pancreatic β-cell-specific Rictor-deficient mice were generated and analyzed [116]. mTORC2 activity in pancreatic β-cells was reduced in Rictor-deficient mice, resulting in the inhibition of serine 473 phosphorylation of AKT. This result induces a decreased pancreatic β-cell volume due to the suppression of pancreatic β-cell proliferation, as well as decreased insulin secretion due to decreased GSIS and insulin content. The phosphorylation of AKT serine 473 by mTORC2 regulates pancreatic β-cell volume and function by translocating PDX1 into the nucleus via FOXO1 phosphorylation. It has also been shown that compensatory changes, such as hyperinsulinemia and increased β-cell mass, are cancelled in Rictor-deficient mice [117]. The loss of these compensatory changes has been shown to be due to the inhibition of PKCα activation by mTORC2.

## 8. Is mTOR “Good” or “Bad” for Pancreatic β-Cells?

As described above, mTOR has diverse roles in pancreatic β-cells, and it regulates the function and quantity of pancreatic β-cells through various signaling pathways. The study of genetically modified mice has advanced our understanding of the role of mTOR, and many transgenic mice related to mTOR signaling have been generated, some of which are described below.

βTSC2KO mice, reported by Shigeyama et al. in 2008, show increased pancreatic β-cell mass, hyperinsulinemia, and hypoglycemia at a young age but decreased pancreatic β-cell mass, hypoinsulinemia, and hyperglycemia in old age [73]. This biphasic change was thought to be caused by a decrease in insulin receptor substrate 2 expression due to the negative feedback associated with the constitutive activation of mTORC1. As a result, there is a decrease in β-cell number probably due to attenuated insulin signaling. In 2014, the constitutive activation of mTORC1 was reported to induce pancreatic β-cell apoptosis by inhibiting autophagy and mitophagy [99]. In the state of mTORC1 activation, mitochondrial DNA transcription is increased by PGC1α activation, and mitochondrial production is increased, but the inhibition of mitophagy prevents the degradation of old mitochondria, resulting in the accumulation of abnormal mitochondria in pancreatic β-cells. This is also considered an important factor for pancreatic β-cell failure. Rachdi et al. independently generated and analyzed pancreatic β-cell-specific TSC2-deficient mice and found increased pancreatic β-cell mass, increased insulin secretion, and improved glucose tolerance [74]. However, unlike the aforementioned βTSC2KO mice, these mice did not show biphasic changes and remained hyperinsulinemic throughout their lives. It is difficult to explain the reason for the phenotypic differences between these two strains of mice, and there are many possibilities. For example, they could be due to differences in the mouse strains, but both were well backcrossed to B6 mice, or due to differences in the constructs of the TSC2-floxed allele or in environmental conditions that affect the phenotype, such as variation in the microbiota in the breeding environment or the amino acid content of the food. Further studies are needed to clarify such details.

Other genetically modified mice related to mTOR include mice lacking pancreatic β-cell-specific TSC1, which forms a complex with TSC2, that showed increased β-cell mass [44]. Furthermore, mice in which mTORC1 is activated by overexpressing Rheb specifically in pancreatic β-cells have increased pancreatic β-cell mass, enhanced insulin secretion, and improved glucose tolerance [79]. These mice do not show biphasic changes. In contrast, mice lacking mTOR or RAPTOR specifically in pancreatic β-cells have decreased pancreatic β-cell mass and hyperglycemia [24,80,82,118]. Furthermore, mice overexpressing kinase-dead mTOR display decreased PDX1 expression in pancreatic islets and decreased insulin secretion, although there is no change in pancreatic β-cell mass [83].

From these reports, it is clear that mTOR plays essential roles in pancreatic β-cells, and the loss or long-term inactivation of mTOR may be an important cause of pancreatic β-cell failure. However, there is a difference of opinion as to whether mTOR activation is beneficial for maintaining pancreatic β-cell mass and blood glucose levels. Recent reports suggest that phosphatases called pleckstrin homology domain leucine-rich repeat protein phosphatases 1 and 2 (PHLPP1/2), the expression of which is upregulated by activated mTORC1, regulate pancreatic β-cell apoptosis [119]. In 2021, Brouwers et al. found that FURIN, a proprotein convertase, is highly expressed in human pancreatic islets, and mice lacking FURIN specifically in β-cells show decreased pancreatic β-cell mass, decreased insulin secretion, and glucose intolerance. The mechanism underlying the decrease in insulin secretion and glucose intolerance is thought to be ATF4 activation caused by increased mTORC1 activity in pancreatic β-cells [84]. In contrast, analysis of Rab1a-knockout mice revealed that amino acid-Rab1a-mTORC1 signaling maintains the identity and insulin secretion of pancreatic β-cells through the expression of PDX1 [87].

## 9. Conclusions

Despite the fact that pancreatic β-cells are the most important contributors to the development and progression of diabetes, there are still many aspects of their biology that remain unknown. In addition to the small amount of tissue and the difficulty in isolating them from humans, this may be due to the complexity of the many signaling pathways that exist in these cells. In particular, nutritional factors, such as glucose, amino acids, and free fatty acids, regulate pancreatic β-cell mass and insulin secretion via the stimulation of many signaling pathways. At the center of these signals is mTOR. Previous reports have revealed the existence of various upstream and downstream signals of mTORC1, all of which are crucial for maintaining glucose homeostasis in pancreatic β-cells. Notably, the role of mTOR may vary depending on specific environments and the degree and duration of its activation. These results are very interesting because they suggest that mTOR may have an impact on the phenotypic diversity of type 2 diabetes. In addition, mTORC1 activity is increased in the islets of patients with type 2 diabetes, suggesting that it also plays a key role in its development in humans [120]. However, it is not clear whether this molecule or signal can be a therapeutic target. There are still many unanswered questions, especially regarding the role of mTORC2 activation and its effect on differentiation, and more research is needed. It is hoped that the elucidation of the role of mTOR in pancreatic β-cells will lead to a complete picture of the fate of pancreatic β-cells.

## Figures and Tables

**Figure 1 biomolecules-12-00614-f001:**
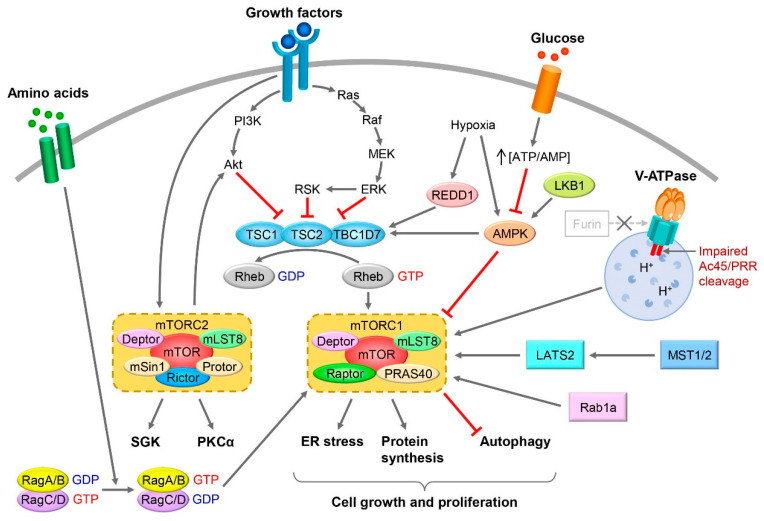
mTORC1 and mTORC2 activation signals in cells. In cells, mTORC1/2 is activated by growth factors, glucose, and amino acids. In particular, mTORC1 positively and negatively regulates cell growth and proliferation through protein synthesis, ER stress, and inhibition of autophagy. Gray arrows indicate activation, line segments in red indicate inactivation. The symbol of cross means that the action of Furin on V-ATPase is inhibited [30].

**Figure 2 biomolecules-12-00614-f002:**
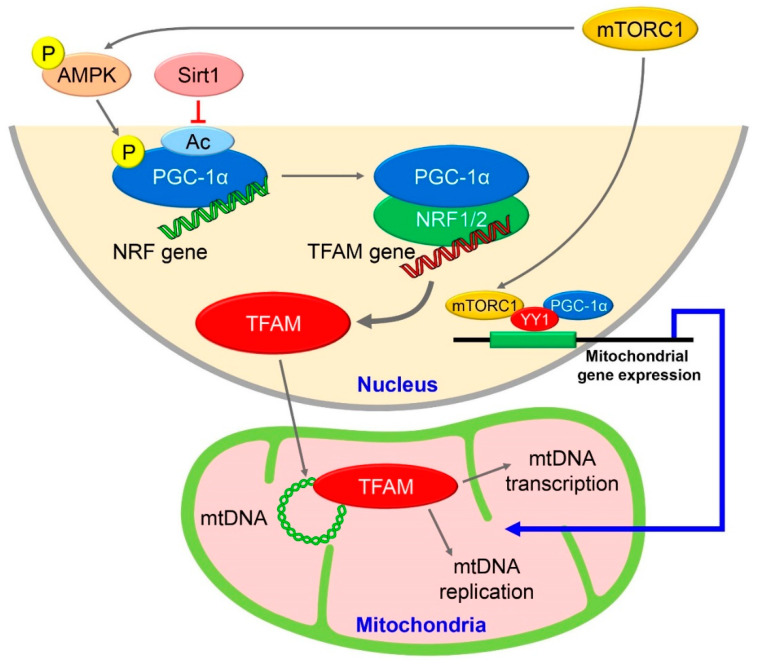
mTORC1 and mitochondrial biogenesis. AMPK, phosphorylated by mTORC1, promotes NRF gene transcription by activating PGC1α, which, in turn, activates mtDNA replication and transcription via TFAM. At the same time, activated mTORC1 contributes to mitochondrial biogenesis by promoting mitochondrial gene expression by binding to PGC1α and YY1 in the nucleus. Gray arrows indicate activation, line segmant in red indicates inactivation. Blue arrows indicate transcribed mitochondrial genes function within mitochondria [65].

**Figure 3 biomolecules-12-00614-f003:**
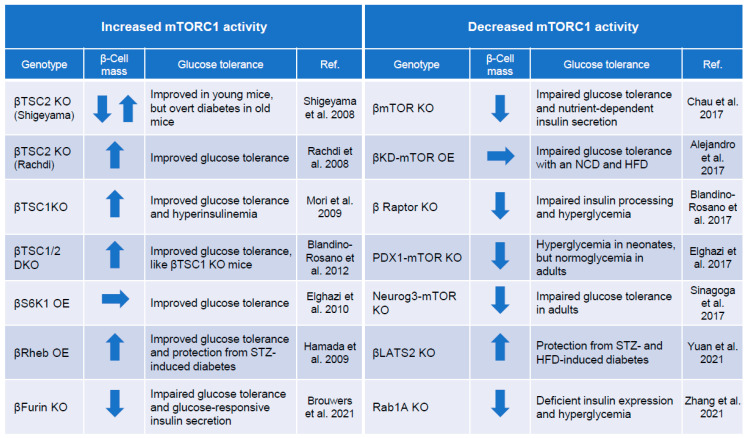
Pancreatic β-cell mass and glucose tolerance in mTORC1-related transgenic mice. In genetically modified mice related to mTORC1 activity in pancreatic β-cells, changes in mTORC1 activity affect pancreatic β-cell mass and glucose tolerance. In this figure, the left panels show mice with increased mTORC1 activity [24,44,73,74,76,79,84], and the right panels show mice with decreased mTORC1 activity [77,78,80,83,85,86,87]. In β-cell mass regulation, up arrows indicate an increase, down arrows indicate a decrease, and horizontal arrows indicate no change. The coexistence of upward and downward arrows indicates change in response to age. DKO: double knockout, HFD: high-fat diet, KD: kinase dead, KO: knockout, NCD: normal chow diet, OE: overexpression, STZ: streptozotocin.

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
