# Peer review of "Roles of mTOR in the Regulation of Pancreatic β-Cell Mass and Insulin Secretion"

_biomolecules, 2022, doi:10.3390/biom12050614_

Round 1

Reviewer 1 Report

Paper by Asahara et al presents an almost exhaustive paramount of mTORC1 role in the function of β-cells and the progression of diabetes, and they suggest that mTOR has both positive and negative effects on pancreatic β-cells for the development of diabetes.

On the other hand an essential point is that the polymorphic effect of mTORC1 on pancreatic β-cells for the development of diabetes is heavily influenced by  different metabolic and environmental conditions, in which mTORC2 pathway activation might play a role. Actually, since mTOR is important for metabolism-related pathologies, understanding the distinct and overlapping regulation and functions of the two mTOR complexes is vital for the development of more effective therapeutic strategies. A discussion of the key discoveries on the regulation and metabolic functions of the mTOR complexes appear unavoidable. As an example a discussion on the current models for how growth factors (GFs), such as insulin, trigger mTORC2 activation, and its role in pancreatic cell proliferation

Author Response

Dear Reviewer1

Thank you for reviewing our manuscript. Your comments and suggestions were very significant to our review.

As you point out, mTORC2 plays an important role in pancreatic beta cell proliferation. That is why I added the chapter 'mTORC2 and β-cell growth and function'. In this chapter, we introduced the mechanism of insulin activation and its function in pancreatic β-cells based on previous reports.

Reviewer 2 Report

Roles of m TOR in the regulation of pancreatic β-cell mass and insulin secretion.

Abstract: Line 19; “-are the only type of cells”.

Introduction: Line 56; “-of m TOR on them”. You do not have to repeat “pancreatic β-cells”. Please note that references to the sources of the cartoons (Fig. 1 & 2) are required within the text unless they are original.

Line 113; A sentence should not begin with “Because”. Line 253; “—showed increased β-cell mass---“.

Recommendation: This is a highly informative, well-written and well-referenced review, which is worthy of publication. The minor corrections indicated should be implemented.

Author Response

Dear Reviewer2

Thank you for reviewing our manuscript. Your comments and suggestions were very significant to our paper.

Abstract: Line 19; “-are the only type of cells”.→ We corrected it.

Introduction: Line 56; “-of m TOR on them”. You do not have to repeat “pancreatic β-cells”.→We corrected it.

Please note that references to the sources of the cartoons (Fig. 1 & 2) are required within the text unless they are original. →Fig.1 is the original, but ref64 has been added as a reference paper for Fig.2.

Line 113; A sentence should not begin with “Because”. Line 253; “—showed increased β-cell mass---“.→We corrected it.